# Sustainable Sesame (*Sesamum indicum* L.) Production through Improved Technology: An Overview of Production, Challenges, and Opportunities in Myanmar

**Daisy Myint [1,2], Syed A. Gilani [3], Makoto Kawase [4] and Kazuo N. Watanabe [5,6,*]**

1    Graduate School of Life and Environmental Sciences, University of Tsukuba, Tsukuba, Ibaraki 305-8572, Japan; s1736030@s.tsukuba.ac.jp

2    Department of Agriculture, Ministry of Agriculture, Livestock and Irrigation, Office No. (43), Nay Pyi Taw 15011, Myanmar

3    Department of Biological Science and Chemistry, University of Nizwa, P.O. Box 33, PC 616, Birkat Al Mouz, Nizwa 616, Oman; gilani@unizwa.edu.om

4    Faculty of Agriculture, Tokyo University of Agriculture, Tokyo 243-0034, Japan; mk207198@nodai.ac.jp

5    Faculty of Life and Environmental Sciences, University of Tsukuba, Tsukuba, Ibaraki 305-8572, Japan

6    Tsukuba-Plant Innovation Research Center, University of Tsukuba, Tsukuba, Ibaraki 305-8572, Japan

*    Correspondence: watanabe.kazuo.fa@u.tsukuba.ac.jp; Tel.: +81-29-853-4663

**Abstract:** This paper aims to review the research achievements concerning sustainable sesame (*Sesamum indicum* L.) production and outlook on the production constraints and future perspectives for Myanmar sesame. Sesame is an economically and nutritionally important crop, and it is prized for oil. The global sesame market demand is rising with increasing health awareness. Meanwhile, there is high competition in the market among producing countries for an international trade. Smallholder farmers in developing countries cultivate sesame as a cash crop on marginal soils. The edible oilseed sectors currently face several challenges, including ones affecting sesame crops. For sustainable production of sesame, an integrated approach is needed to overcome these challenges and the critical limiting factors should be identified. In recent years, sesame genomic resources, including molecular markers, genetic maps, genome sequences, and online functional databases, are available for sesame genetic improvement programs. Since ancient times, sesame has been cultivated in Myanmar, but productivity is still lower than that of other sesame producing countries. Myanmar sesame production is limited by many factors, including production technology, research and development, etc. With integration of these genomic resources, crop production and protection techniques, postharvest practices, crop improvement programs, and capacity building will play a crucial role for improving sesame production in Myanmar.

**Keywords:** oilseed; *Sesamum indicum* L.; advanced breeding technologies; genomic resources production constraints; opportunities; strategy; Myanmar

## 1. Introduction

Sesame (*Sesamum indicum* L.) is labeled as the queen of oilseeds because of its high oil content, delicious nutty aroma, and flavor [1] and is traditionally categorized as a health food in Asian countries [2]. Sesame seed is used for a wide array of edible products in raw or roasted formand also for industrial uses such as soaps, lubricants, lamp oil, an ingredient in cosmetics; pharmaceutical uses, and animal feed [3]. It contains a considerable amount of oil, proteins, carbohydrates, essential

minerals, a high amount of methionine and tryptophan, fibers as well as secondary metabolites such as lignans, saponins, flavonoids, and phenolic compounds. Moreover, the seeds are a good source of calcium, phosphorus, and iron and are rich in vitamin B, E, and a small amount of trace elements. Sesame oil has a pleasant, mild taste and is remarkably stable. It has a high content of polyunsaturated fatty acids, oleic, and linoleic acid. Sesame oil has an excellent stability due to natural antioxidants, i.e., sesamin, sesamolin, and sesamol [4,5].

Worldwide sesame seed consumption was USD 6559.0 million in 2018, and it will reach USD 7244.9 million by 2024, with a CAGR (compound annual growth rate) of 1.7% [6]. Global sesame consumption is steadily increasing mainly due to changing consumer's consumption patterns and increasing health awareness. Nowadays, the consumers mostly prefer the high nutritive value products. Consequently, the demand of sesame seeds is higher since it has several nutritional characteristics such as vitamins, minerals, fiber, healthy fat, and protein. About 70% of the world's sesame seed is used to produce oil and meal. Total annual oil and food consumption is about 65% and 35%, respectively [7]. Tanzania is the highest sesame seed consuming country (21% based on tonnes), followed by China (19%), Sudan (9%), Myanmar, India, Ethiopia, and Nigeria (6% each) with almost 74% of the global consumption. The consumption of Tanzania, Sudan, and Myanmar was 30.8, 17.6, and 10.1 kg per year respectively in 2016 [8]. The world sesame production is about 5,532,000 metric tons (MT) behind soybean, groundnut, cottonseed, sunflower, linseed, and rapeseed, in the quantity of world oilseed production. The average sesame productivity of the world's top producing countries within 20 years (1999–2018) is given in Figure 1. However, the data for Sudan and South Sudan is officially recorded in 2012 by Food and Agriculture Organization Statistical Databases. India, Myanmar, and China are the highest producers among the countries. Average sesame yield is found to be highest in China (1223 kgha$^{-1}$) followed by Nigeria (729 kgha$^{-1}$ and Tanzania (720 kgha$^{-1}$) [9].

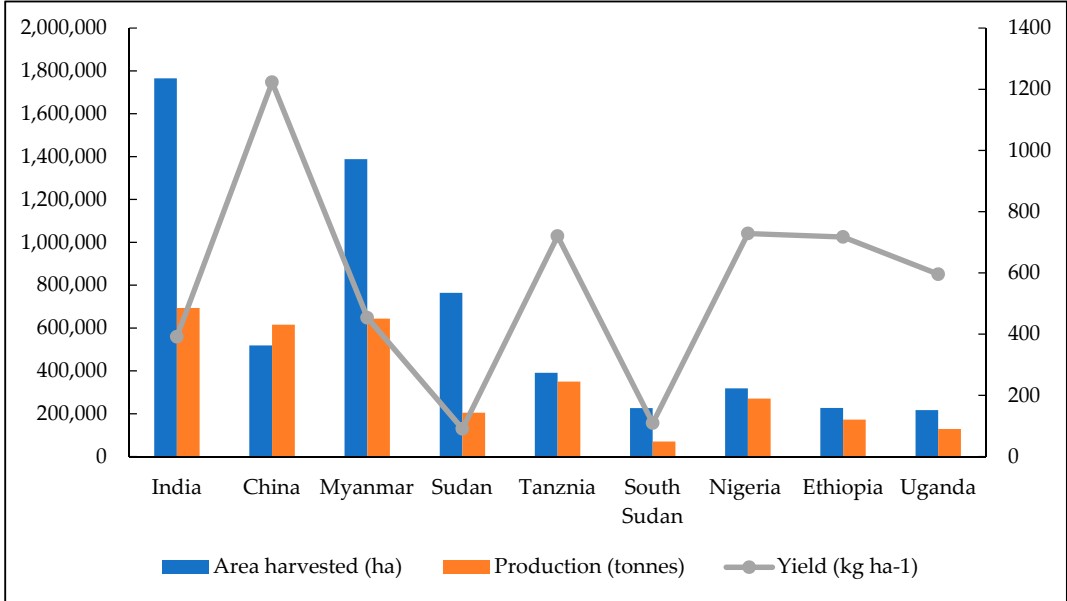

**Figure 1.** Trend of sesame production values in top producing countries during the last twenty years (Average 1999–2018) Source: Food and Agriculture Organization Statistical Databases (FAOSTAT), 2020 [9].

In 2018, 6,016,000 MT were grown worldwide on 11,743,000 hectares (ha) with an average yield of 512 kgha$^{-1}$ (Table 1). Asia and Africa produce nearly 97% of the world's supply of sesame. Sudan devotes the greatest acreage but has one of the lower records for yield per hectare. Tanzania produces nearly 14.6% of the world's sesame crop, followed by Myanmar at 12.78% and India at 12.4% (Table 1).

**Table 1.** Area, production, and yield of sesame in the selected countries and the world in 2018.

| | Area ('000 ha) | Yield (kgha$^{-1}$) | Production ('000 MTha$^{-1}$) | % of World Production |
|---|---|---|---|---|
| **Top producing countries** | | | | |
| India | 1730 | 431 | 746 | 12.40 |
| China | 311 | 1393 | 433 | 7.20 |
| Myanmar | 1463 | 525 | 769 | 12.78 |
| Sudan | 3480 | 282 | 981 | 9.33 |
| Tanzania | 800 | 701 | 561 | 14.56 |
| South Sudan | 618 | 334 | 207 | 3.43 |
| Nigeria | 539 | 1063 | 573 | 9.52 |
| Ethiopia | 415 | 726 | 301 | 5.01 |
| Uganda | 210 | 667 | 140 | 2.33 |
| **Regions** | | | | |
| Asia | 3906 | 578 | 2257 | 37.52 |
| Africa | 7549 | 474 | 3575 | 59.42 |
| America | 288 | 636 | 183 | 3.04 |
| Europe | 1 | 975 | 1 | 0.02 |
| World | 11,743 | 512 | 6016 | - |

Source: Food and Agriculture Organization Statistical Databases (FAOSTAT), 2020 [9].

Myanmar is one of the top ten sesame producing countries and ranks the third largest in cultivated area and the second largest in production. Sesame is an economically important crop not only for producing edible oil but also for domestic and international markets. It is also an essential component for Myanmar cultural ceremonies and traditional food. It also serves as cooking oil, a garnish, a snack, and a flavoring agent in some foods. Recent production statistics indicate a steady increase in sesame production, but it is still mostly traditional. Compared to other sesame producing countries, the sesame yield is meager, and local oil sufficiency is still not enough. Thus, annual imports of palm oil are considered as sufficient for local oil consumption. Export of good-quality sesame seeds from the country and selling low–standard-quality seeds in the local market for the domestic oil industry results inedible oil with poor quality and low recovery. According to the Myanmar Edible Oil Dealer's Association, Myanmar's annual edible oil consumption has risen to over one million tons due to population growth and food business development. Consequently, under consumer's health awareness, high-quality edible oil is imported from Malaysia and Indonesia.

It is predicted that vegetable oil consumption will be doubled, and sesame oil consumption may be 100 million MT by 2030 [10]. Therefore, the demand for genetic studies of oil-rich crops will increase to improve oil content. Sesame is one of the oldest oilseeds and is widely cultivated in both tropical and subtropical areas [11]. It is grown on marginal land by small and medium farmers in those areas. The edible oilseed sectors currently face several challenges, including ones affecting sesame crops. Although Myanmar is one of the top producers and has a vast diversity of sesame germplasm, it still faces numerous constraints for sesame production. Sesame cultivation is limited due to low and unstable yield and the strategies based on technology are also limited. Nowadays, considerable research accomplishments as well as genetic resources and genome sequence information are available for the sesame crop improvement program. The country's current status and research and development is needed for the development of the sesame oilseed sector. Therefore, the purpose of this paper is to review research achievements, current trends, and challenges in sesame production. It further aims to propose future opportunities and the strategy for an improvement of Myanmar's sesame production. It is conducted by using the secondary data and literature search.

## 2. Major Constraints on Sustainable Sesame Production

Sesame thrives well in a harsh environment and requires limited fertilizer, water, and litter without the need for the use of pesticides due to high levels of natural tolerance for diseases and insects. However, the yield is highly variable depending on the growing environment, cultural practices, and the cultivars. It is mostly grown under rainfed conditions of arid and semi-arid areas where mild-to-severe water deficit stress is experienced. Sesame productivity is limited in those areas by drought and salinity. It is sensitive to drought mainly at the vegetative stages [12] in all of its growing regions and has low production potential in semiarid regions due to drought stress. Grain yield as well as oil yield and quality are decreased depending on genotypes and drought intensity. Sesame cannot tolerate salinity it is especially sensitive to excessive calcium and sodium chlorides ions in soil solution. Several studies have shown that sesame is tolerant to high salinity levels at germination and initial growth stages, and variability has been found between sesame genotypes [13].

Sesame is sensitive to waterlogging, salinity, and chilling that limit sustainable production [14]. Sesame growth and yield decreases after 2–3 days of waterlogging when the crop is grown on poorly drained soils. Waterlogging significantly reduces plant growth, leaf axils per plant, seed yield, and net photosynthesis [15]. A variety of insect pests attack the seedling, foliage, flowers, pods, and stem of sesame. These are the primary causes of yield reduction and an average loss of 25% of potential worldwide production [16]. Leaf spot, stem, and leaf blotch, and *Cercospora* leaf spots are bacterial diseases that mostly damage the sesame. Wilt is also devastating on susceptible varieties. Additionally, blight, charcoal rot, stem anthracnose, mildew, and phyllody are significant diseases in sesame [16]. Similarly, the crop may drastically be affected by insect pests such as leaf roller, capsule borer, sphinx moth, aphids, and gall midge.

Lack of fast-adapting cultivars, capsule shattering, uneven ripening, poor crop stand establishment, lower fertilizer responses, profuse branching, low harvest index, indeterminate growth habit, and susceptibility to diseases are the limiting factors in sesame production worldwide [14]. The indeterminate growth habit and the shattering nature of sesame cause harvesting problems and result in yield loss and poor adaptation to mechanized harvesting. The majority of the world's sesame (probably over 99%) is shattering, and most of the harvest is manual. Harvesting practices vary from country to country and from one place to another within countries. The sesame plants are harvested when they have 50% maturity. The stalks are tied into small bundles, then stacked to dry, threshed either on the floor or on plastic/cloth in the field to collect the seed. Its capsule shattering nature is the most problematic issue because of high seed losses (up to 50%) at harvesting time [17]. This character is not suitable for mechanized harvesting [18] and limited for commercial production in countries that have no available labor.

Postharvest loss is the loss of grain between the moments of harvest and consumption that occurs at all stages of postharvest handling: processing, transportation, storage, packaging, and marketing. The major determinant sources of sesame postharvest losses were farm size, total sesame grain produced, weather conditions, distance when piles are transported, stacking days, the distance of the sesame farm, and mode of grain transportation [19]. High production and postharvest losses are also challenges [20]. The serious loss in quantity and quality that occur in oilseeds is mainly due to the adoption of improper postharvest technologies. This leads to the incidence of damaged, discolored, shriveled, and malodorous kernels in the product.

## 3. Research Achievements through Improved Technologies in Sesame

The global population will grow by 25% and reach 10 billion over the next 30 years, and high-yielding, more nutritious, pest and disease resistant, and climate-smart new crops need to develop [21]. A study on oilseed crops' competitiveness highlighted seven strategic topics and pointed out the key research fields of oilseed genetic improvement: nutritional balance related to oil quality, meal quality for animal feeding, production systems, environmental challenges, and non-food uses from vegetable oils [22]. Primarily, sesame breeds should be selected for increased yield, increased oil

content, uniformity, and biotic and abiotic stress resistance. Although sesame has high value and is essential for dietary uses, health benefits, and industrial applications, attention on genetic improvement is still lagging behind compared to that for other oilseed crops [23]. Since the yield gap is enormous, the breeding target is to increase the yield potential.

Yield-related traits such as the number of capsules per plant and growth habit are more considered. Langham discovered a monogenic homozygous recessive indehiscent mutant (*id/id*) in 1943 [24]. The indehiscent type has disseminated through the world, and most breeders started to make the crossing with other sesame lines. In 1944, mechanized harvesting in sesame was initiated in Venezuela. In Bulgaria, as a result of long-time research (30 years), four sesame varieties, namely Victoria, Aida, Valya, and Nevena were successfully produced for mechanized harvesting along with the average yield of 1354 kg per hectare by collaboration between breeders and engineers [25].

The genus *Sesamum* includes 23 species [26], and *S. indicum* is well known and widely cultivated. It is adapted to a wide range of environments, and it has a large diversity. India, China, Central Asia, the Middle East, and Ethiopia have been proposed as five centers for sesame genetic diversity [27]. A huge amount of genetic materials of cultivated and related wild sesame are currently preserved in gene banks of India, China, South Korea, the United States, and also small-scale gene banks in some Asian and African countries. Sesame core collections have already been done for efficient exploration and utilization of novel genetic variation [28] and resulted in 362 Indian accessions [29], 453 Chinese accessions [30], and 278 Korean accessions [31]. They are valuable genetic resources for current and future sesame genetic improvement.

Genetic engineering techniques overcome the shortcomings of conventional breeding, and various innovative approaches apply for sesame breeding. Sesame's recalcitrant nature hinders the application of modern biotechnology [32]. Somatic embryogenesis was successful from hypocotyl segments [33], cotyledons, root, and subapical hypocotyl of seedlings [34]. Moreover, several researchers have attempted to produce callus tissues from different methods and media [14]. Recently, the regeneration protocol from cotyledons [35] and de-embryonated cotyledons of sesame [36] was refined. Furthermore, the effective micropropagation system for the conservation and multiplication of sesame has been updated. This will be helpful for genetic transformation, reproductive growth, and other tissue culture studies [37]. *Agrobacterium*-mediated genetic transformation of sesame has been reported, but the transformation frequency is low [38,39]. Recently, a high-frequency transformation technique for sesame resulted in high regeneration and a transformation frequency of 57.33% and 42.66%, respectively [40].

Current crop breeding strategies will not provide a sufficient supply to meet the demands for food and nutritional security for the ever-increasing population. 5G breeding strategies: Genome assembly, Germplasm characterization, Gene function identification, Genomic breeding methodologies, and Gene Editing technologies are proposed to accelerate crop genetic improvement [39]. Genome assembly provides genomic tools and techniques for trait discovery and molecular breeding. It is vital to create a gene expression atlas, proteome maps, metabolome maps, and epigenome maps [41]. Scientists from Oil Crops Research Institute of the Chinese Academy of Agricultural Science and other institutes have successfully generated a high-quality sesame genome. Genome assemblies from two landraces (*S.indicum*cv. Baizhima and Mishuozhima) and three modern cultivars (*S.indicum* var. Zhongzhi 13, Yuzhi 11, and Swetha) are now available, and it furnishes a substantial resource for comparative genomic analyses and gene discovery.

The Sesame Genome Working Group produced a 293.7Mb draft assembly of modern cultivar, Yuzhi11 [42]. It has a small diploid genome (~357Mb) and the draft assembly consisted of 274 Mb in 16 linkage groups and contained 27,148 predicted protein-coding genes [43]. The draft genome assemblies for two landraces Baizhima and Mishuozhima originating from Hainan and Zhejiang provinces were produced in China [44]. The genome assembly of Swetha, an elite modern cultivar from India, was created by a team from the National Bureau of Plant Genetic Resources, resulting in the largest assembly to date of 340Mb [45]. Recently, the sesame pangenome was constructed

from two landraces and three modern cultivars. It resulted in a 554.05 Mbp genome with core and dispensable genomes of 258.79and 295.26 Mbp, respectively. It consists of 26,472 orthologous genes clusters, of which 58.21% were core and 15,890 were variety-specific genes. The study found that modern cultivars from China and India display significant genomic variation. Modern varieties contain genes mainly related to yield and quality, while the landraces contain genes involved in environmental adaptation. The sesame pangenome provides a resource for further sesame crop improvement [46]. Table 2 presents the online database of sesame genomic sequence information. They are valuable tools for functional and molecular breeding of sesame.

**Table 2.** List of online databases for sesame genomics.

| Database Name | Website | Ref. |
|---|---|---|
| The sesame genome project | http://www.sesamegenome.org | [47] |
| Sinbase | http://www.ocri-genomics.org/Sinbase/index.html | [48] |
| SesameHapMap | http://202.127.18.228/SesameHapMap/ | [44] |
| PMDBase | http://www.sesame-bioinfo.org/PMDBase | [49] |
| SesameFG | http://www.ncgr.ac.cn/SesameFG | [50] |
| Sesame Germplasm Resource Information Database | http://www.sesame-bioinfo.org/phenotype/index.html | - |
| ocsESTdb | http://www.ocri-genomics.org/ocsESTdb/index.html | [51] |
| PTGBase | http://www.ocri-genomics.org/PTGBase/index.html | [52] |
| SisatBase | http://www.sesame-bioinfo.org/SisatBase/ | - |

Understanding the genetic variability, heritability, and correlation studies of plant traits plays a crucial role in the effective use of germplasm in any breeding program. National and International genebanks provide a rich source of the diverse allele that is vital for crop improvement. DNA markers are also powerful tools for genetic evolution, marker-assisted breeding of crops, and they accelerate modern plant breeding because of enhancing the genetic gain and reducing breeding cycles in many crop species [23]. A combination of molecular markers should be applied to study the genetic diversity of indigenous and exotic germplasm.

Several workers developed and applied different types of molecular markers for sesame including random amplified polymorphic DNA (RAPD) [53], amplified fragment length polymorphism (AFLP) [54], simple sequence repeat (SSR) types: Inter-simple sequence repeats (ISSR) [55], expressed sequence tags SSR [56–58], cDNA-SSR [59,60], genome sequence SSR [49,61–63], chloroplast SSR [57], high-throughput methods for SNPs (single-nucleotide polymorphisms), including restriction site-associated DNA sequencing (RAD) [64,65], specific length amplified fragment sequencing (SLAF) [47], RNA- seq [62], whole-genome sequencing [44,48,66], genotyping by sequencing [67], insertion–deletions (Indels) [62,64]. Many of these markers have been using for genetic diversity, molecular breeding, and genetic mapping. Online sesame databases provide valuable information related to molecular functions, genome components, gene expression, SSR, SNP, QTL (quantitative trait locus) and functional genes, transposons, genetic maps (Table 3). These are valuable sources for the Myanmar sesame improvement breeding program.

**Table 3.** Reference quantitative trait loci (QTLs) and gene names of target traits for sesame.

| Traits | QTLs/Genes | Markers Type | Marker Numbers | Mapping Population | Parent of the Cross | Ref. |
|---|---|---|---|---|---|---|
| **Production enhancement** | | | | | | |
| Grain yield | *Qgn-1*, *Qgn-6*, | SLAF | 9378 | 150 BC1 | Yuzhi 4 × Bengal Small-seed | [68] [47] [65] |
| Grain number per capsule | *Qgn-12* | | | | | |
| Thousand grain weight | *Qtgw-11* | | | | | |
| Seed coat color | QTL-1, QTL11-1, QTL11-2, QTL13-1 | | | | | |
| Seed coat color | *qSCa-8.2*, *qSCb-4.1*, *qSCb-8.1*, *qSCb-11.1*, *qSCl-4.1*, *qSCl-8.1*, *qSCl-11.1*, *qSCa-4.1* and *qSCa-8.1* | SLAF SNP | 1233 | 107 F2 430 Recombinant inbred lines (RILS, F8) | Zhongzhi No.13 ×Jiaxiang Sesame Zhongzhi No.13 × Semi-dwarf ZZM 2748 | |
| Seed coat color | *SiPPO* (*SIN_1016759*) | SSR | 400 | 500 RILs (F6) | Zhongzhi 13 × Mishuozhima | [69] |
| Plant height | *Qph-6* and *Qph-12* | SNP | 1,800,000 | 705 worldwide accessions | | [44] |
| Semi-dwarf sesame plant phenotype | QTL (*qPH-3.3)*, Gene [*SiGA20ox1*(*SIN_1002659*)] | SNP SSR | 400 | 430 RILS (F8) 500 RILs (F6) | Zhongzhi No.13 × Semi-dwarf ZZM 2748 Zhongzhi 13 × Mishuozhima | [65] [69] |
| Plant height | *SiDFL1* (*SIN_1014512*) and*SiILR1* (*SIN_1018135*) | SNP | 1,800,000 | 705 worldwide accessions | | [44] |
| Capsule number per plant | *Qcn-11* | SNP SSR InDels | 1190 22 18 | 224 (RIL), F8:9 | Miaoqianzhima × Zhongzhi 14 | [64] |
| First capsule height | *Qfch-4*, *Qfch-11*, and *Qfch-12* | | | | | |
| Capsule axis length | *Qcal-5* and *Qcal-9* | | | | | |
| Capsule length | *Qcl-3*, *Qcl-4*, *Qcl-7*, *Qcl-8*, and *Qcl-12* | | | | | |
| Number of capsules per axil | *SiACS* (*SIN_1006338*) | SNP | 1,800,000 | 705 worldwide accessions | | [44] |
| Mono flower vs. triple flower | *SiFA* | SLAF (Marker58311, Marker34507, Marker36337) | 9378 | 150 BC1 | Yuzhi-4 × Bengal Small-seed | [68] |
| Flowering time | *SiDOG1* (*SIN_1022538*) and *SiIAA14* (*SIN_1021838*) | SNP | | 705 sesame accessions | | [44] |
| Determinate trait in sesame | gene *SiDt* (*DS899s00170.023*) | SNP | 30,193 | 120 F2 | Yuzhi 11 (indeterminate, Dt) × Yuzhi DS 899 (determinate dt1) | [66] |

**Table 3.** *Cont.*

| Traits | QTLs/Genes | Markers Type | Marker Numbers | Mapping Population | Parent of the Cross | Ref. |
|---|---|---|---|---|---|---|
| Branching habit | *SiBH* | SLAF (Marker129539, Marker41538, Marker31462) | 9378 | 150 BC$_1$ | Yuzhi-4 × Bengal Small- seed | [68] |
| Recessive GMS | recessive GMS gene*SiMs1* | AFLP markers *P01MC08*, *P06MG04*, *P12EA14* | | 237 NILs (Near-Isogenic Lines) | Sib mating between 95ms-5A and 95ms-5B | [70] |
| dominant GMS gene*Ms* | *SBM298* and *GB50* | SSR | 1500 | Noval GMS line W1098A (Backcrossing and sib-mating; BC$_2$F$_6$ | Conventional variety Ezhi 1 × wild germplasm Yezhi2 | [71] |
| **Stress related** | | | | | | |
| Water logging tolerance | *qEZ09ZCL13*, *qWH09CHL15*, *qEZ10ZCL07*, *qWH10ZCL09*, *qEZ10CHL07*, and *qWH10CHL09* | SSR (*ZM428*) closely linked to*qWH10CHL09* | 113 | 206 RIL F$_6$ | Zhongzhi No.13 (high tolerance to waterlogging) × Yiyangbai (extremely sensitive to waterlogging) | [72,73] |
| Drought tolerance | TFs (Transcription Factors) families (AP2/ERF and HSF) | - | - | - | - | [74,75] |
| Drought, salinity, oxidative stresses, charcoal rot | Osmotin-like gene (*SindOLP*) | - | - | - | - | [76] |
| **Gene for Oil traits** | | | | | | |
| Sesamin production | *SiDIR (SIN_1015471)*, *SiPSS (SIN_1025734)* | SNP | 1,800,000 | 705 worldwide accessions | | [44] |
| Oil content | *SIN_1003248, SIN_1013005, SIN_1019167, SIN_1009923 SiPPO (SIN_1016759) SiNST1 (SIN_1005755)* | | | | | |
| Fatty Acid composition | *SiKASI (SIN_1001803), SiKASII (SIN_1024652), SiACNA (SIN_1005440), SiDGAT2 (SIN_1019256), SiFATA (SIN_1024296), SiFATB (SIN_1022133), SiSAD (SIN_1008977), SiFAD2 (SIN_1009785)* | | | | | |
| Sesamin and sesamolin content | *SiNST1 (SIN_1005755)* | | | | | |
| Protein content | *SiPPO (SIN_1016759)* | | | | | |

SLAF: specific length amplified fragment sequencing; SNP: single nucleotide polymorphism; SSR: simple sequence repeat; AFLP: amplified fragment length polymorphism; Indels: insertion–deletions; GMS: genetic mate sterility

## 4. Overview of Myanmar Sesame Production

### 4.1. Past and Present Status of Sesame Production in Myanmar

Myanmar is primarily an agrarian country with three main agro-ecological zones: Deltaic area, Central DryZone (CDZ), and Hilly areas. The country is composed of 14 states and regions and the Nay Pyi Taw Union Territory. Approximately 18.6% of total land area (678,500 sq. kilometers) is devoted to agriculture, and69.32% of livelihoods depend directly or indirectly on this sector. The agricultural products are the second-largest export commodity, which contributes 25.6% of GDP and 24.4% of the total export earnings. Cereal crops are the major crops that account for 40% of the total cultivated area. Oil seeds are the third largest grown crop after pulses and the second most important food crop after rice. Myanmar has optimal conditions for the production of edible oil seeds being rich in diverse agro-ecological zones. Major oilseed crops are groundnut, sesame, sunflower, mustard, and niger. Among them, sesame is the largest cultivated crop and occupies 51% of the total oilseed crops cultivation (Figure 2).

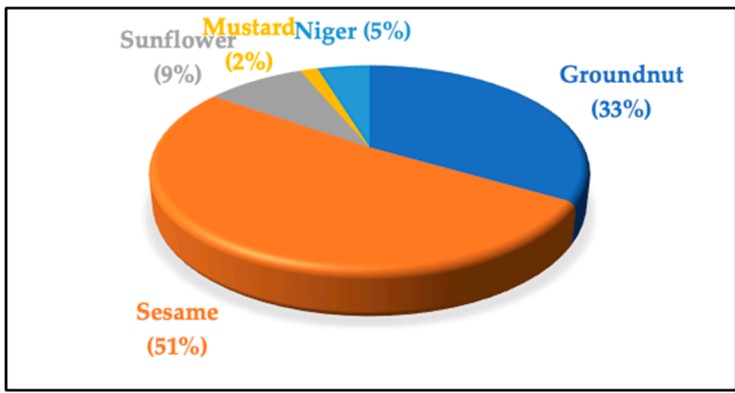

**Figure 2.** Share of oilseed crops in 2017–2018, source: Ministry of Agriculture, Livestock and Irrigation (MOALI), 2018 [77].

Sesame ("Hnan" in Burmese) has been cultivated since ancient times. Sesame has been cultivated in 1044 AD, the time of the Bagan Dynasty [78]. According to the government report, approximately 0.17 million hectares of sesame was cultivated in 1894–1895. Table S1 in the Supplementary Materials describes sesame sown area, yield, and production of different eras; colonial Era (1900–1948), Independent Era (1948–1949 to 1973–1974), Socialist Era (1974–1975 to 1987–1988), State Peace and Development Council (1988–1989 to 1999–2000). In 1900–1901, the cultivated area was 335 thousand hectares, and yield data were recorded in 1914–1915. In 1947–1948, the planted area was 559 thousand hectares, which was not very much different from that in 1914–15 with a sown area of 490 thousand hectares. However, yield decreased to less than half (307 to 109 kgha$^{-1}$) due to long-time production on the same land and world war. In the Independent era, the sesame sown area increased nearly double from 1948 to 1974, and the yield went up. In the Socialist period, sesame production was dramatically improved with higher yield because irrigated sesame began to be cultivated, and four sesame varieties were released.

From 1989–1990 to 1999–2000, intensive sesame cultivation zones, including irrigated sesame, were carried out to attain self-sufficiency in edible oil. The production was increased by expanding area and using high-yielding varieties; Sinyadanar-4 (white) and Sinyadanar-3 (black) sesame. The cultivated sown area, production, and yield from 2000–2018 is shown in Figure 3. Between 2000 and 2008, the production and yield increased with the value of production 376 thousand tons to 840 thousand tons and with the value of yield 303 to 587 kgha$^{-1}$. In 2009, the production and yield significantly dropped because of severe drought in that year. Afterward, total production increased, but it was not much different within seven years. In 2017–2018, the production and yield declined as the rainfall was erratic.

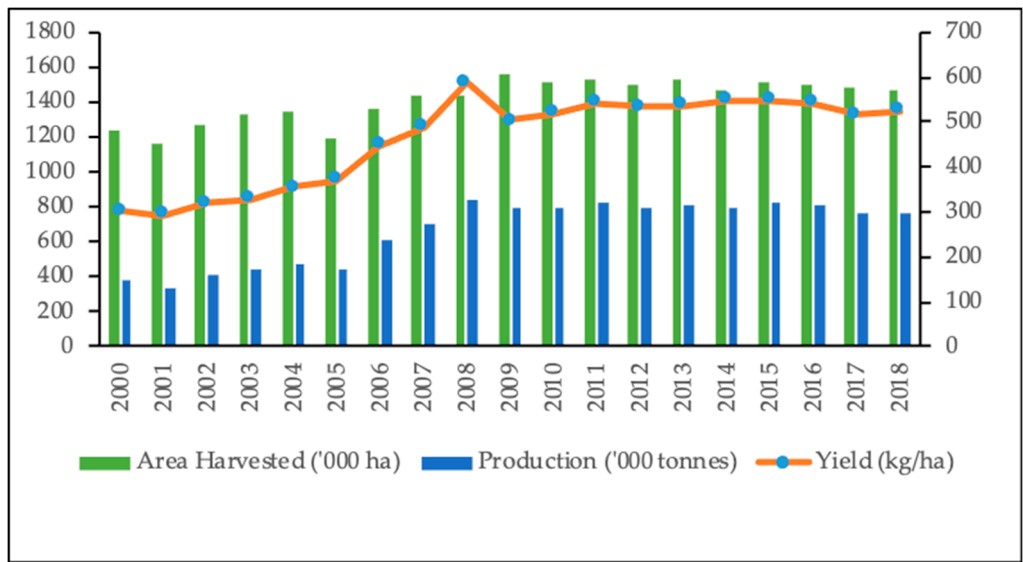

**Figure 3.** Sesame sown area, production, and yield between 2000 and 2018, source: Ministry of Agriculture, Livestock and Irrigation (MOALI), 2018 [77].

Production takes place exclusively in Mandalay, Magway, and Sagaing Regions located in the central dry zone of Myanmar. The climatic conditions of those areas favor the cultivation of edible oil crops and it is also known as "Myanmar's Oil Bowl". Sesame can be planted and harvested throughout the year. However, the primary growing season is the monsoon and a small number of crops is cultivated during the winter season and summer season. Monsoon sesame is planted in May and harvested in August. In the winter season, it is cultivated in September or October and harvested in December. The cultivated area in regions and states of Myanmar and production distribution is presented in Figure 4a,b.

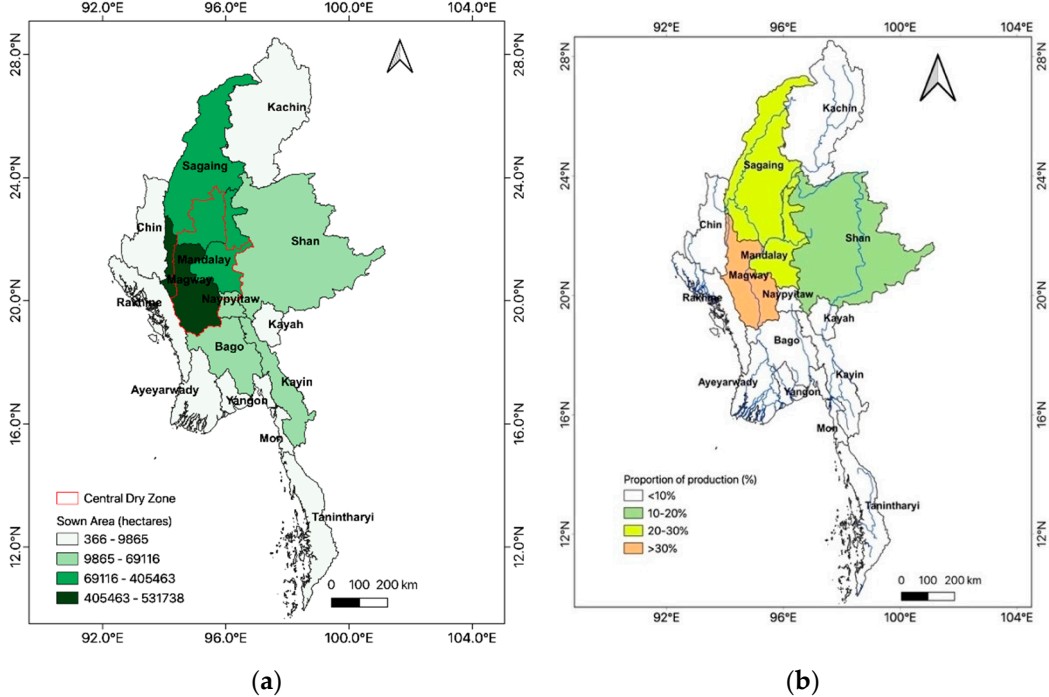

**Figure 4.** (**a**) Sesame sown area in regions and states of Myanmar, (**b**) distribution of sesame production in Myanmar (data based on last 3 years' average production). Source: authors' construction.

### 4.2. Sesame Genetic Resources and Research Activities

Myanmar is home to many tropical crops species and a high rate of diversity is found in sesame [79]. Since 1945, Myanmar has started the collection of sesame landraces; over 200 cultivars including, traditional varieties, cultivars developed through research and trials, and over 100 varieties imported from Japan, Australia, America, Mexico, etc. Crossbreeding is practiced between traditional ones and imported varieties. Although a variety's name varies from region to region, Myanmar's sesame name is related to the stem, capsule, and seed color. Principally, they can be grouped as early (60–80 days), medium (80–100 days), and late maturity (100–135 days) on the basis of life period. The early matured varieties are Pat-Le-War-Hmyaung, She-Ka-lay, Pin-Lon-War, Bok-hmwe, Man-Shwe-Wa, Mai-Thi-hla, and Ye-Kyaw. Medium maturity varieties are Pa-De-Tha, Tha-Tun-Phyu, Magaw red sesame (25/160), Gyu× Ni ×Pok, Gwa-tayar, Bok-Pyik, Mei-Daw-Let The, Hnan-Wa-Gale, Magway 2/21, Magway 7/9. The late maturity varieties are Kye-Ma-Shaung and Red Magway 50/2 [80]. The varieties existing in Myanmar are presented with their basic characteristics in Table S2 in the Supplementary Materials.

Sesame hybridization between traditional and exotic varieties started since 1954. The collection of traditional cultivars, selection, crossbreeding, research and development, and the distribution of quality seed is being undertaken [80]. There is scanty research activity in sesame and a lack of information about genetic diversity has been a barrier to improve sesame in Myanmar. So far, the genetic diversity study based on morphology showed a wide range of variability for stem, leaf, and flower traits, and yield and yield components character. The yield contributing factors are days to first flowering, days to 50% flowering, plant height, number of primary branches per plant, number of secondary branches per plant, number of capsules per plant, and capsule length. Ten clusters were obtained, and they are helpful in a breeding program [81]. Ten sesame varieties have been screened against seedling blight and stem and root rot. The author suggested that MR 9A variety could be applied in crop production as parent material in disease resistance breeding [82]. Myanmar sesame phyllody is studied based on 16S ribosomal DNA sequence analysis. It is reported that the phytoplasma association with phyllody is related to '*Candidatus Phytoplasma asteris*' belonging to 16SrI-B in Myanmar [83].

### 4.3. Myanmar's Sesame Export and Quality

Myanmar produces the three types of sesame seeds, namely white, black, and mixed. The white one is commonly roasted and used as a topping in bakery products and snacks and salads, and it has a reasonable market price for domestic and export. Black is a high-value ingredient exported to Japan and other consuming countries, and the mixed seeds (red, brown, yellow) are used for oil extraction and medicinal purposes [80]. Myanmar contributes 9.5% in global sesame export. The main importer of sesame is China, which shares 74.2% of total sesame export and other importers are South Korea, Japan, Singapore, Thailand, and Hongkong. Raw sesame products go directly to China through the Muse exchange center, Myanmar–China border area to China by wholesalers and Chinese commission agents, and to Japan and Taiwan through Yangon exporters. Roasted sesame powders sell to Korea via Yangon port [84].

Since 1991–1992, Myanmar exported about 48,775 tons of sesame seeds with a value of US\$ 33.84 million (Table 4). Table 4 indicates the trend of sesame seed and oil export quantity and export values during 2010–2017.

**Table 4.** Sesame seed and oil export values of Myanmar.

| Years | Sesame Seed | | Sesame Oil | |
|---|---|---|---|---|
| | Export Quantity (tons) | Export Value (US$) | Export Quantity (tons) | Export Value (US$) |
| 2010 | 53,700 | 53,348 | 800 | 450 |
| 2011 | 36,500 | 30,000 | 800 | 470 |
| 2012 | 38,200 | 37,500 | 800 | 480 |
| 2013 | 33,300 | 32,000 | 800 | 500 |
| 2014 | 25,679 | 44,220 | 161 | 631 |
| 2015 | 19,814 | 31,944 | 65 | 292 |
| 2016 | 24,506 | 38,410 | 44 | 190 |
| 2017 | 33,500 | 47,447 | 13 | 88 |

Source: Food and Agriculture Organization Statistical Databases (FAOSTAT), 2020 [9].

Generally, the export quantities were lower between 2010 and 2015. After 2015, sesame seed export rose sharply in 2017, and most of them are exported to China (90%), Japan (6%), and Singapore (3%) in Asia and Switzerland in Europe (nearly 300 tons). However, the sesame oil export trend gradually declined. Since the potential for further exports is high, endeavors need to be initiated by upgrading the oil industry. According to the Myanmar Ministry of Commerce, the export quantity increased by about 123,100 tons in 2018–2019 [9].

Food safety issues are of particular importance for international trade for sesame seeds and its products. A common requirement for importing countries is that the crop is produced under Global GAP (Good Agricultural Practice) standards. Market standardizations, grades, and requirements depend on countries that used the end-products. Myanmar has standards and technical regulations, as well as SPS (Sanitary and Phytosanitary Measures) mainly based on an international standard such as food standards, which are adopted from CODEX. Standard specifications and GAP guidelines are used to inspect exported and imported agricultural products. Myanmar's sesame seeds have the specification for export; seed purity (98%–99%), oil content (48%–51%), free fatty acid (FFA) (2% Max), admixture (1%), other color mixed (2% Max), moisture (8% Max), clean, dry, and no mold.

*4.4. Constraints and Challenges of Myanmar's Sesame Production*

The major challenge in sesame production is to increase the productivity per unit area. Despite its production increase, the sesame yield has been stagnant for many years at around 0.5 tons per hectare. According to FAOSTAT, Myanmar's sesame yield (525 kgha$^{-1}$) is significantly less than that of neighboring China (1393 kgha$^{-1}$). In sesame production, the gap between the potential yield and the farmer's yield is due to various abiotic, biotic, technological, and socioeconomic factors. Myanmar is ranked second in the list of countries most affected by climate change, and the average annual temperature is expected to rise over the next century, with the magnitude of warming varying by region and season. During the 2020s, the national yearly average temperatures are projected to rise 0.7–1.1 °C compared with that in the 1980–2005. By the 2050s, the dry regions will experience a substantial percentage change in temperature and rainfall [85]. A short monsoon season reduces the sesame yield [84], and heavy rain also causes quantity and quality losses in sesame. If there was rain during stalk drying, it took a long duration to dry for threshing and caused high losses [85]. Excessive rainfall was the most frequent reason for yield loss for upland sesame and green gram, according to 18% of upland sesame growers.

Low agricultural productivity is the result of multiple factors, many of which are associated with the undersupply of qualified farm inputs. The government distributes the sesame seed about 307 MT that includes 89 MT of certified seeds [86]. However, the availability of quality seeds is insufficient

and most of the farmers in the Magway region are using their own seed harvested from the previous crop season [86,87]. There are minimal seed production activities in cooperation with private seed companies. Pest and disease attack is one of the limiting factors in sesame production. Aphids, thrips, sesame jassids (*Orosius albicinctus* Distant), sesame leaf rollers (*Antigastra catalaunalis*), and leaf hoppers were severe pests in the sesame cultivation regions. Phyllody, charcoal rot, *Alternaria* blight, *Cercospora*, bacterial leaf spot, and leaf roll are prominent diseases in sesame [88]. It is necessary to know the incidence and distribution of each sesame disease and pest and its impact on production to manage it in the field correctly. Screening tests within local varieties and strains should be conducted site specifically to select the resistance to a local pest and diseases.

Land degradation is one of the factors threatening the livelihoods of dry zone farmers and causes limitations for sesame production because the soil characteristics are clay and sandy soils with a high risk of water shortage and erosion [89]. The soil has low fertility, low humus content, poor water storage capacity, and high evaporation. Crop yields of monsoon rice, groundnut, sesame, and cotton in the highly degraded area were found to be 3–12 times lower than those in a less degraded area. Farmers in highly degraded areas faced crop yield reduction, increased cultivation cost, and increased uncultivable land area [90]. Therefore, effective measures and practices for soil conservation are required, and strengthening activities of farmers' awareness through extension services need to be implemented.

Poor postharvest management is another barrier for sesame production. Farmer awareness and the technical support on postharvest loss is still limited. Most of the postharvest loss studies conducted focused on cereal crops and vegetables, and the study of sesame loss is scanty in Myanmar. So far, the average loss for stalk drying and threshing was 4.77% if there was rain during the stalk drying periods in some farmers' fields. It takes a longer duration for stalk drying before threshing and causes high losses. The average winnowing loss was 0.28%, and the total postharvest losses on the farm before storage were on average 6.1%. In Myanmar, sesame is grown by small and medium farmers and they harvest sesame manually, which is a highly labor-intensive operation. Currently, labor shortage is another problem faced from production, to drying, to harvesting. It can be overcome by introducing appropriate mechanical harvesting devices. Since oilseeds are semi-perishable, they are subject to serious quality losses during storage. Unless they are correctly stored, microbial proliferation, insect and rodent infestation, and biochemical changes can happen. Storage losses were the greatest among the whole postharvest losses of sesame. It can be mitigated by the use of efficient storage technology, upgrading infrastructure, and better storage practices [86].

Lower production and productivity in some areas is due to a lack of market. Still, it is attributable to the need for better crop management and technical expertise, reflective of the low level of research and development and extension activities [20]. Efficient research strategies may be required to reduce the effects of various yield reducing factors in sesame production. Against this background, the following figure provides a simplified strength, weaknesses, opportunities, and threats (SWOT) analysis of Myanmar's sesame production and value chain [80,84,87,91]. Most of the farmers want to use good-quality seed, chemical fertilizers, and pesticides with reasonable prices. Moreover, they are faced with the uncertainty or low-quality inputs. Furthermore, an insufficient amount of credit to use the inputs for sesame is another cause of low yield. In the sesame value chain, the various actors such as the wholesaler, millers, processors, and exporter received the high demand for both local and international trades. However, they are also facing many challenges; the highest investment rate for wholesalers, the highest interest rate for processors, uncompetitive with exporters in buying sesame seeds, and poor-quality seeds for oil milling for millers, and no access to test pesticide residues, especially imidacloprid. Generally, the major constraints were financial problems for them, and this constraint should be solved concomitantly for all actors along the sesame value chain (Figure 5).

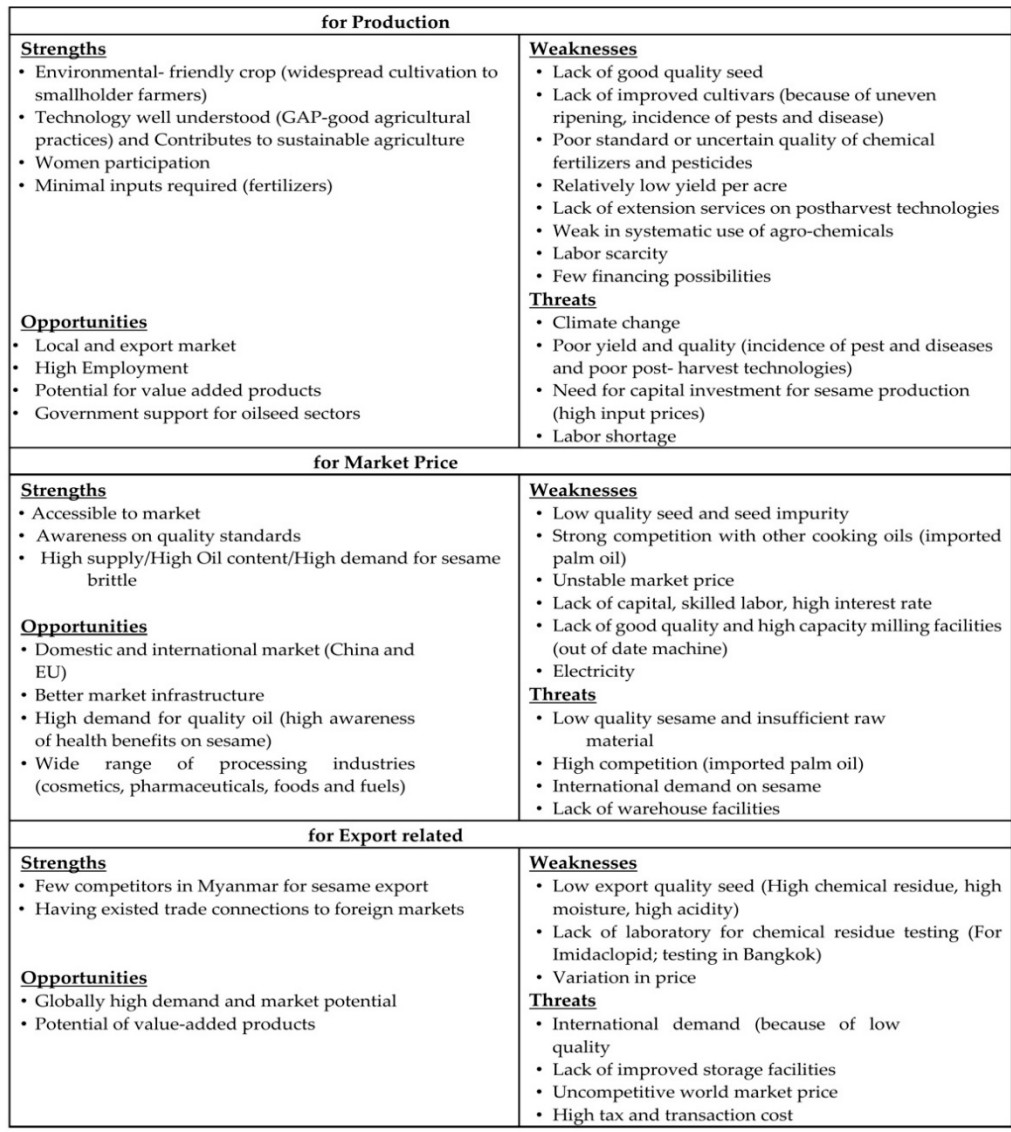

**Figure 5.** Strengths, weaknesses, opportunities, and threats (SWOT) analysis for sesame production and value chain.

## 5. Future Opportunities, Vision, and Strategies Proposed for Enhancing Sesame Production

Due to increasing health awareness, people are more concerned about nutrition and the quality of sesame produce. Besides the improvement of yield and increased resistance to external stress, the way to improve the quality and efficiency is by focusing on the market demand. In sesame breeding, quality breeding is much more valuable than yield breeding [92]. Global industrialization and modernization are causing continuous increased high demand, and the energy crisis is increasing day by day. Fossil fuels have many adverse effects on human health and the environment; thus, the utilization of renewable and sustainable energy may be a fruitful solution. Many researchers have focused on biodiesel as a renewable energy resource because of its potential. Biodiesel can be produced from edible and non-edible vegetable oils, and fuel from vegetable oil emits less pollution than diesel fuels [93]. Biodiesel production is expanding worldwide due to energy security and environmental concerns. Sesame is a potential oilseed crop for biodiesel production [94–99].

Myanmar's Agricultural Policy vision focuses on an inclusive, competitive, food and nutrition secure, climate change-resilient, and sustainable agricultural system that contributes to the socioeconomic well-being of farmers and rural people and the further development of the national economy. The agricultural development strategy will seek the inclusive value chain development to

achieve this policy [100]. Government policy is to achieve self-sufficiency in edible oil and maintain the oil price, to solve the challenges and constraints, to get higher bargaining power, and attract the world market. All stakeholders, including government organizations, researchers, private sectors, and farmers, need to be have collaborated efforts. Especially, introduction and identification of potential and adaptable better cultivars, development of high-yielding varieties with good quality traits by modern breeding techniques, improved agronomic and management practices, strengthening and capacity building of postharvest handling process, and upgrading the oil processing facilities. The most important recommendation is to enhance the collaborative partnership between national and international sesame teams so that the production barriers can be addressed, and effective breeding strategies can be implemented.

Myanmar has suitable environmental conditions for sesame production and proximity to the international market. The extraction of oil from oil crop seed is another highly profitable industry. Hence, the development of the oil extraction industry will benefit not only the rural farming sector but also the national economy. It has been supposed that if farmers put quality seeds and input with GAP, then collectors and processors can get more raw materials at competitive prices; they can invest in their oil mills, packaging, equipment, and on value-adding processing. On the other hand, if exporters can get more high-value seeds, they can invest in state-of-art private logistic infrastructure, edible export oil, and exploit the international market requirements. Therefore, edible oil sectors need to upgrade by getting higher production [90].

Currently, the yield is above the world's average level, and oil quality is relatively good, but it needs to improve further. Most of the farmers are still struggling to meet the high standards required by the foreign markets because of poor agronomic practices, and weather-related crop failures result in low yields and large pre- and postharvest losses. Development of high-yielding varieties with better quality and with non-shattering traits, and tolerance to biotic and abiotic stress should be encouraged to increase the sesame productivity. Although there is a variety of development and agronomic research by the Department of Agricultural Research (DAR), it is not enough to bring increased productivity. Promising sesame landraces are available in Myanmar, but it is a further need to identify elite lines that are conducive to the currently prevailing condition. Local germplasm assessment and evaluation should be done in collaboration with the national genebank. Seed and oil yield of the germplasm should be enhanced genetically by using conventional and modern techniques, including QTL and the use of Marker Assisted Selection (MAS).

Higher yields can be obtained using improved varieties with better management of soil, water, insects, pests, and diseases. Similarly, better nutrient management increases sesame productivity. Research and extension are necessary to expand the use of Integrated Pest Management for sesame, where intensive pesticide application is practiced. Since sesame is grown by marginal farmers in rainfed conditions, it is crucial to assure irrigation facilities. Timely availability of quality seed through the strengthening of the seed industry and other inputs, fertilizer, and pesticides at an affordable price needs to increase. The government needs to provide foundation seeds for seed multiplication. Public and private seed companies need to multiply certified seeds derived from the foundation seeds by encouraging farmers' participation. The labor shortage in harvesting is one of the major problems due to rural–urban migration. Farm mechanization will help to minimize the production cost by saving labor costs. The following programs are proposed for enhancing sesame productivity in Myanmar.

1. Genetic enhancement and conservation of local germplasm
2. Institutionalization of quality seed production and distribution mechanisms
3. Development of varieties with suitable traits (high oil content, tolerant to drought, waterlogging, phyllody, and sesame black stem rot)
4. Development of area-specific production technology packages (integrated crop, soil, and pest management)
5. Strong coordination and linkages with research, extension, and private organizations for the effective implementation of scaling up the oilseed production technology

6.　　Improved specialized laboratory testing for residues, enhancing awareness training on postharvest management and pesticides use, farm machinery, and quality assurance systems to meet the international standards.

**Supplementary Materials:** The following are available online at http://www.mdpi.com/2071-1050/12/9/3515/s1, Table S1: Sesame sown area, harvested area, yield, and production from 1900 to 2001 and from 1999 to 2000. Table S2: Basic characteristics of sesame varieties released in Myanmar.

**Author Contributions:** Writing—original draft preparation, D.M.; writing—review and editing, S.A.G. and K.N.W.; information collection, M.K.; supervision, K.N.W.; funding acquisition, K.N.W. All authors have read and agreed to the published version of the manuscript.

**Funding:** This paper was supported in part by JSPS grant-in-aid #17H01682 and by Plant Transgenic Design Initiative (PTraD) at the University of Tsukuba, Japan.

**Acknowledgments:** D.M thanks the Ministry of Education, Culture, Sports, Science, and Technology (MEXT), Government of Japan, for a supporting scholarship and the Department of Agriculture, Ministry of Agriculture, Livestock, and Irrigation of Myanmar for sharing information.

**Conflicts of Interest:** The authors declare no conflict of interest.

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
