# Peer review of "Sustainable Sesame (Sesamum indicum L.) Production through Improved Technology: An Overview of Production, Challenges, and Opportunities in Myanmar"

_sustainability, doi:10.3390/su12093515_

Round 1
Reviewer 1 Report
Dear Authors,
I revised the manuscript "Sustainable Sesame (Sesamum indicum L.) Production through Improved Technology: An Overview of Production, Challenges and Opportunities in Myanmar" submitted to the Sustainability Journal. The manuscript is interesting. However, I have some concerns, which need to be addressed before considering for final publication.
The entire paper needs close copy editing for English language style and grammar.
Check units format - use the SI units.
Table 1. FOASTAT 2020 – need reference number.
Line 132. Check "quality and quality".
Line 179. Check " at the end of the line?
Line 386. Figure 4 is too small.
Author Response
|
1. |
The entire paper needs close copy editing for English language style and grammar. |
English language style and grammar has been edited accordingly. |
|
2. |
Check units format- use the SI units.
|
SI unit system is used now. |
|
3. |
Table-1. FAOSTAT 2020- need reference number |
Reference No. as [9] has been incorporate on Line 82 |
|
4. |
Line 132. Check “quality and quality” |
It has been corrected as “quantity and quality” on Line 149. |
|
5. |
Line 179. Check ” at the end of the line? |
The sentence has been edited on Line 195 |
|
6. |
Line 386. Figure is too small |
Figure 4 has been redrawn with large size (Line 424)
|
Reviewer 2 Report
This review focuses on the overviewing the research achievements concerning the sustainable sesame (Sesamum indicum L.) production and outlook on the production constraints and future perspectives for Myanmar sesame. The manuscript is very interesting and the idea is nice. The title is clear and it is adequate to the content of the article. The author’s work on discussing achieved results is appreciated. The minor revisions are necessary to improve the clarity of the presentation. I have some recommendations for authors:
- Moderate English changes required;
- Please specify the review working method.
- Please improve the quality of figure 4.
Author Response
|
Moderate English changes required |
English grammar and style has been changed accordingly. |
|
Please specify the review working method. |
The reviewing method is added (Line 107-Line 108). |
|
Please improve the quality of figure 4. |
Figure 4 has been redrawn with large size (Line 285)
|
Reviewer 3 Report
I found this review interesting for the community working with Sesame
there are some suggestions below
1) L59, data of 2016 are reported while at L61 there are data of 2018. Why the authors don't show a graph with the trend of the last 10 or 20 years in the main producer countries. These data can easily findable on FAOSTAT
2) In table 3 i would give more details, adding the mapping population (BC, F2, RIL etc) used for QTL mapping, the parent of cross, the number of individuals, the type and the number of molecular markers. This information will be highly appreciated by readers and make more useful the paragraph L208-L219
3) Figure 4 is not readable, I would report only the main information while the rest as a part written in the text
Author Response
|
L59, data of 2016 are reported while at L61 there are data of 2018.
|
Line 59 is supported for seed consumption and L61 is for production data. |
|
Why the authors don’t show a graph with the trend of the last 10 0r 20 years in the main producer countries. These data can easily findable on FAOSTAT. |
Yes, a summary figure as “Figure 1” has been incorporated (Line 67-Line 71). The information is presented in Line 62-Line -66.
|
|
In table 3, I would give more details, adding the mapping population (BC,F2,RIL etc) used for QTL mapping, the parent of cross, the number of individuals, the type and the number of molecular markers. This information will be highly appreciated by readers and make more useful the paragraph L208-L219. |
The information has been incorporated in Table 3(Line 241). |
|
Figure 4 is not readable, I would report only the main information while the rest as a part written in the text. |
Figure 4 has been redrawn and the information is described in Line 416-Line 425. |
Round 2
Reviewer 3 Report
All comments have been raised accordingly